# Naphthyridine Derivatives Induce Programmed Cell Death in *Naegleria fowleri*

**DOI:** 10.3390/ph14101013

**Published:** 2021-10-01

**Authors:** Aitor Rizo-Liendo, Iñigo Arberas-Jiménez, Endika Martin-Encinas, Ines Sifaoui, María Reyes-Batlle, Javier Chao-Pellicer, Concepción Alonso, Francisco Palacios, José E. Piñero, Jacob Lorenzo-Morales

**Affiliations:** 1Instituto Universitario de Enfermedades Tropicales y Salud Pública de Canarias, Universidad de La Laguna, Avda, Astrofísico Fco. Sánchez, S/N, 38203 La Laguna, Spain; arizolie@ull.edu.es (A.R.-L.); iarberas@ull.edu.es (I.A.-J.); isifaoui@ull.edu.es (I.S.); mreyesba@ull.edu.es (M.R.-B.); alu0101016429@ull.edu.es (J.C.-P.); 2Departamento de Obstetricia, Ginecología, Pediatría, Medicina Preventiva y Salud Pública, Toxicología, Medicina Legal y Forense y Parasitología, Universidad de La Laguna, 38203 La Laguna, Spain; 3Departamento de Química Orgánica I, Facultad de Farmacia and Centro de Investigación Lascaray (Lascaray Research Center), Universidad del País Vasco/Euskal Herriko Unibertsitatea (UPV/EHU), Paseo de la Universidad 7, 01006 Vitoria-Gasteiz, Spain; endika@ipna.csic.es (E.M.-E.); concepcion.alonso@ehu.eus (C.A.); francisco.palacios@ehu.eus (F.P.); 4Red de Investigación Colaborativa en Enfermedades Tropicales (RICET), 28006 Madrid, Spain; 5Consorcio Centro de Investigacion Biomedica En Red M.P. (CIBER) de Enfermedades Infecciosas, Inst. de Salud Carlos III, 28006 Madrid, Spain

**Keywords:** *Naegleria fowleri*, meningoencephalitis, naphthyridine derivatives, programmed cell death

## Abstract

Primary amoebic encephalitis (PAM) caused by the opportunistic pathogen *Naegleria fowleri* is characterized as a rapid and lethal infection of the brain which ends in the death of the patient in more than 90% of the reported cases. This amoeba thrives in warm water bodies and causes infection after individuals perform risky activities such as splashing or diving, mostly in non-treated water bodies such as lakes and ponds. Moreover, the infection progresses very fast and no fully effective molecules have currently been found to treat PAM. In this study, naphthyridines fused with chromenes or chromenones previously synthetized by the group were tested in vitro against the trophozoite stage of two strains of *N. fowleri*. In addition, the most active molecule was evaluated in order to check the induction of programmed cell death (PCD) in the treated amoebae. Compound **3** showed good anti-*Naegleria* activity (61.45 ± 5.27 and 76.61 ± 10.84 µM, respectively) against the two different strains (ATCC^®^ 30808 and ATCC^®^ 30215) and a good selectivity compared to the cytotoxicity values (>300 µM). In addition, it was able to induce PCD, causing DNA condensation, damage at the cellular membrane, reduction in mitochondrial membrane potential and ATP levels, and ROS generation. Hence, naphthyridines fused with chromenes or chromenones could be potential therapeutic agents against PAM in the near future.

## 1. Introduction

Primary amoebic meningoencephalitis (PAM), caused by infection from the free-living amoeba parasite *Naegleria fowleri*, is a fatal central nervous disease that exhibits severe meningitis and cranial pressure caused by inflammation of the brain [1]. This species, commonly known as the “brain-eating amoeba”, is the only species of the *Naegleria* genus associated with the infection of humans [2], mostly affecting healthy children and young adults [3].

After the first reported case described by Carter and Fowler in 1965 [4], more than 430 cases have been successfully reported worldwide [5]. The USA is the country with the most cases reported at 156 cases [6], followed by Pakistan with 105 cases described until 2018 [6,7]. The incidence of reported cases of PAM has increased in recent years, and it is still considered underdiagnosed due to the non-specific symptoms [8,9].

The infection is produced after the performance of water activities in warm water bodies that present *Naegleria*. The trophozoites penetrate the olfactory nerve via the nasal cavity. Once there, *Naegleria* migrate to the cribriform plate, cross it, and invade the central nervous system [10]. The unspecific symptomatology includes seizures, stiff neck, bifrontal headache or paralysis and appears within the first 9 days after exposure [11]. PAM cases present a fast progression, with the death of the patient taking place between 1 and 18 days after the appearance of the first symptom and a reported mortality rate of above 95% in the reported cases [12,13].

Along with this fast progression, the lack of rapid clinical detection methods [14,15,16] and the absence of specific therapy protocols make it even more difficult to face PAM disease, showing a correlation with the high mortality rate reported [1]. Current experimental therapy options involve miltefosine or amphotericin B, sometimes administrated in combination with other medication (for example, with azoles or rifampin) or with controlled hypothermia [17,18]. Despite the announcement of a successful PAM treatment case [19,20,21], these compounds present undesired side effects such as nephrotoxicity or even brain damage [22,23]. Hence, there is an urgent need to develop novel anti-*Naegleria* treatments more efficiently with low cytotoxicity effects for the patient.

Coumarins and 2*H*-chromenes (Figure 1) are heterocyclic derivatives with a benzene fused to a pyran [24,25,26]. They are present in a large number of natural products, and both heterocycles constitute an important moiety in bioactive heterocycles with pharmacological properties [27], including anticancer drugs [27,28]. Therefore, these scaffolds are very interesting substrates in the search for new drug candidates. The low toxicity of natural products containing chromene and their broad pharmacological properties are attractive features for medicinal chemists and a source of inspiration for the design of novel therapeutic agents [27]. 

The naphthyridine (Figure 1) ring has gained attention, becoming an important pharmacophore in the discovery of new biological agents due to its attractive biological activities [29]. In this sense, several naphthyridine derivatives present good pharmacological activity with low toxicity [30]. In addition, among the most cited applications related to 1,8-naphthyridines are antiparasitic [31,32], antibacterial [33,34], and antiviral [35,36,37] activities.

Furthermore, naphthyridines fused with chromenes or chromenones present diverse biological activities such as antifungal, antimicrobial, and antibacterial [38].

In recent years, our collaborating group has developed the synthesis of fused 1,5 naphthyridine derivatives such as chromeno 4,3-b 1,5 naphthyridine (**I**) and chromenone 4,3-b 1,5 naphthyridine (**II**) and fused 1,8 naphthyridine derivatives such as chromeno 4,3-b 1,8 naphthyridine (**III**) and chromenone 4,3-b 1,8 naphthyridine (**IV**). After studying these compounds as topoisomerase I inhibitors and as antiproliferative agents, it was observed that some of the compounds presented high enzymatic activity; however, no relevant results were obtained in terms of antiproliferative activity [39].

With these considerations in mind, the anti-amoebic activity of a library of 24 fused 1,5 naphthyridine derivatives was tested against *Naegleria fowleri*. Moreover, further studies were carried out in the selected compounds in order to evaluate the induction of programmed cell death (PCD).

## 2. Results

### 2.1. In Vitro Evaluation of Amoebicidal Activity and Cytotoxicity of Fused 1,5 Naphthyridine Derivatives

Among the selected compounds, 1,8 naphthyridines fused with chromene (Figure 2, **1** to **4**) and chromenone rings (Figure 2, **5** to **10**) and 1,5 naphthyridines fused with chromene (Figure 2, **11** to **18**) and chromenone rings (Figure 2, **19** to **24**) were evaluated.

Activity assays revealed that one of the evaluated compounds was able to eliminate *Naegleria fowleri* trophozoites at low µM concentrations, shown in Table 1. Both compounds stood out for their inhibitory concentration 50 (IC_50_) value (Figure 3 and Appendix A), being more selective than miltefosine (one of the reference drugs) [40].

Moreover, compound **3** (IUPAC nomenclature: (6aR, 7R, 12aR)-2-fluoro-7-phenyl-6a,7,12,12a-tetrahydro-6H-chromeno 4,3-b 1,8 naphthyridine) was selected to continue with PCD induction assays due to its good activity (IC_50_) and low toxicity.

### 2.2. Evaluation of PCD Induction by the Selected Compounds

These assays were performed in both strains of *N**. fowleri* and revealed similar results in all PCD experiments.

#### 2.2.1. Double-Stain Assay for the Detection of Chromatin Condensation (Hoechst 33342/PI)

Cells treated with IC_90_ compound **3** revealed chromatin condensation after 24 h of incubation, as shown in Figure 4 and Appendix A with the characteristically bright blue fluorescence.

#### 2.2.2. Plasma Membrane Permeability (SYTOX Green)

Plasmatic membrane damage was caused by the selected compound after IC_90_ incubation for 24 h. Intense green fluorescence could be observed inside cells (Figure 5 and Appendix A).

#### 2.2.3. Generation of Intracellular Reactive Oxygen Species (ROS)

Compound **3** induced the production of ROS in the *Naegleria fowleri* trophozoites (Figure 6 and Appendix A).

#### 2.2.4. Analysis of Mitochondrial Function Disruption

The evaluated compound disrupted the mitochondrial membrane potential in treated cells, which is shown as green fluorescence (Figure 7 and Appendix A) corresponding to the monomeric form of JC-1 dye. To confirm mitochondrial damage, the ATP production level was evaluated (Figure 8).

## 3. Discussion

In this study, the activity of different 1,5 naphthyridine and 1,8 naphthyridine derivatives against the brain-eating amoeba *Naegleria fowleri* was evaluated. Moreover, some of these molecules have already shown good inhibitory antiparasitic activity against other protozoa such as *Leishmania* [41] as well as antiproliferative activity against different cancer cell lines [39,42]. Compound **3** showed good anti-*Naegleria* activity against two different strains (ATCC^®^ 30808 and ATCC^®^ 30215) and a good selectivity compared to the cytotoxicity values against a murine macrophage cell line (Table 1).

On the other hand, Cárdenas-Zuñiga and colleagues described a PCD/apoptosis-like process in *Naegleria fowleri* and the metabolic events that these cells present. Among them, the presence of blebs, DNA condensation, ROS, and electron-dense granules was observed by the authors [43].

Our laboratory is not only focused on the search for new active molecules against the deadly *Naegleria fowleri* but is also concerned with the avoidance of non-desirable side effects that could result in inefficient management of the disease. Compound **3**-treated amoebae showed chromatin condensation, plasma membrane damage, increased levels of ROS, and mitochondrial dysfunction after 24 h, all of which are compatible with the induction of the PCD process. This type of cell death, which avoids the induction of inflammatory reactions that are characteristic of necrosis [44], is the most desirable one in the search for anti-amoebic compounds which present lower side effects and a safer profile in therapy protocols.

In addition, these compounds have previously been described as topoisomerase I inhibitors [39,42]. The genome data of *Naegleria fowleri* have recently been published [45,46], showing the presence of DNA topoisomerase 1-like gene (NF0087810 (https://www.uniprot.org/uniprot/A0A6A5BG47 (accessed on 28 September 2021)) [47]. Therefore, the inhibition of one of the key proteins for the viability of the cell could be the reason for the anti-*Naegleria* activity of the evaluated compounds. Nevertheless, further studies should be developed in order to check cellular targets affected by the evaluated compound **3**. 

Moreover, naphthyridines and their derivatives have been previously reported to inhibit acetylcholinesterases and have been proposed as interesting anti-Alzheimer lead compounds. Hence, these molecules should be able to cross the brain–blood barrier, supporting their potential as anti-PAM agents. However, BBB crossing by these compounds and their in vivo efficacy should be further verified in experimental animal studies in the near future [48].

## 4. Materials and Methods

### 4.1. Amoebic Strains and Cell Line Maintenance

Two American Type Culture Collection strains of *Naegleria fowleri*, ATCC 30808 and ATCC 30215, were used to perform in vitro evaluation of the compounds (LG Promochem, Barcelona, Spain). Amoebae were axenically grown at 37 °C in 2% (*v*/*w*) Bacto Casitone medium (Thermo Fisher Scientific, Madrid, Spain) supplemented with 10% (*v*/*v*) fetal bovine serum (FBS) (Biowest, VWR, Barcelona, Spain), 0.3 μg/mL of Penicillin G Sodium Salt (Sigma-Aldrich, Madrid, Spain), and 0.5 mg/mL of streptomycin sulphate (Sigma-Aldrich, Madrid, Spain). *Naegleria fowleri* cultures were conserved in a biological security facility of level 3 at the Instituto Universitario de Enfermedades Tropicales y Salud Pública de Canarias following the Spanish government’s biosafety guidelines for this pathogen with the minimum number of passages possible to avoid loss of pathogenicity.

Cytotoxicity assays were performed with a murine macrophage cell line (ATCC^®^ TIB-67) grown in Dulbecco’s Modified Eagle’s medium (GIBCO, Thermo Fisher Scientific) (DMEM, *w*/*v*) supplemented with 10% (*v*/*v*) FBS and 10 μg/mL of gentamicin (Sigma-Aldrich, Madrid, Spain). Cells were cultured in 5% CO_2_.

### 4.2. Chemistry General Experimental Information

All reagents from commercial suppliers were used without further purification. All solvents were freshly distilled before use from appropriate drying agents. All other reagents were recrystallized or distilled when necessary. Reactions were performed under a dry nitrogen atmosphere. Analytical thin layer chromatography (TLC) was performed with silica gel 60 F_254_ plates. Visualization was accomplished by UV light. Column chromatography was carried out using silica gel 60 (230–400 mesh ASTM). Melting points were determined with an Electrothermal IA9100 Digital Melting Point Apparatus without correction. NMR spectra were obtained on Bruker Avance 400 MHz and Varian VXR 300 MHz spectrometers and recorded at 25 °C. Chemical shifts for ^1^H NMR spectra are reported in ppm downfield from TMS, chemical shifts for ^13^C NMR spectra are recorded in ppm relative to internal deuterated chloroform (δ = 77.2 ppm for ^13^C), and chemical shifts for ^19^F NMR are reported in ppm downfield from fluorotrichloromethane (CFCl_3_). Coupling constants (*J*) are reported in Hertz. The terms m, s, d, t, and q refer to multiplet, singlet, doublet, triplet, and quartet, respectively. ^13^C NMR and ^19^F NMR were broadband decoupled from hydrogen nuclei. High-resolution mass spectra (HRMS) were measured by EI method with an Agilent LC-Q-TOF-MS 6520 spectrometer. 

### 4.3. Compound Purity Analysis

All synthesized compounds were analyzed by HPLC to determine their purity. The analyses were performed on an Agilent 1260 infinity HPLC system (C-18 column, Hypersil, BDS, 5 μm, 0.4 × 25 mm). All the tested compounds were dissolved in dichloromethane, and 1 μL of the sample was loaded onto the column. Ethanol and heptane were used as mobile phases, and the flow rate was set at 1.0 mL/min. The maximal absorbance at the range of 190−625 nm was used as the detection wavelength. The purity of all the tested compounds was >95%, which met the purity requirement of the journal.

### 4.4. In Vitro Amoebicidal Activity

A colorimetric assay based on the alamarBlue^®^ reagent was used to evaluate the inhibitory concentration 50 (IC_50_) of the compound. Briefly, 50 µL of *Naegleria fowleri* cells from a 2 × 10^5^ cells/mL stock solution was seeded in each well of a 96-well microtiter plate (Thermo Fisher Scientific). After that, 50 µL of serial dilutions of the compound (in the same *Naegleria* culture medium) was added to each well. The negative control consisted of cells with the culture medium alone. Finally, 10% of the final volume of alamarBlue^®^ reagent was placed into the plate. After incubation for 48 h at 37 °C under slight agitation, plates were analyzed with an Enspire^®^ Multimode Plate Reader (Perkin Elmer, Madrid, Spain) using a wavelength of 570 nm and a reference wavelength of 630 nm.

The data were analyzed using GraphPad Prism 9.0.0 software program (GraphPad Software, San Diego, CA, USA). The anti-amoebic activity was expressed as an IC_50_ (inhibitory concentration 50) value which was calculated by performing a nonlinear regression analysis with a 95% confidence limit. All experiments were carried out in triplicate and mean values were calculated. A paired two-tailed *t*-test was used for analysis of the data. Values of *p* < 0.05 were considered significant.

### 4.5. In Vitro Cytotoxicity 

The CC_50_ (cytotoxic concentration 50) of the compound was evaluated using a murine macrophage cell line (ATCC^®^ TIB-67), as previously described [11]. Briefly, cells were incubated for 4 h with different concentrations of compound **3** at 37 °C in a 5% CO_2_ atmosphere.

### 4.6. Analysis of Programmed Cell Death (PCD) Induction

For the evaluation of programmed cell death induction by compound **3** in *Naegleria fowleri*, the ATCC^®^ 30808™ strain was used. Amoebae were incubated with the IC_90_ (inhibitory concentration 90%) of the compound to later evaluate the presence of some of the PCD characteristic metabolic events [49] with different stains, following the manufacturer’s instructions.

#### 4.6.1. Detection of Chromatin Condensation

A double-stain apoptosis detection kit (Hoechst 33342/Propidium Iodide (PI)) (Life Technologies) was used in this assay. Briefly, 5 × 10^5^ cells/mL were incubated with the IC_90_ of the compound for 24 h at 37 °C in a 96-well plate. Images were obtained after 15 min of incubation of the dye using an EVOS™ M5000 Imaging System (Invitrogen by Thermo Fisher Scientific).

This kit allows the differentiation of three type of cells in the culture. Those with red and low blue fluorescence correspond to dead cells (as PI is able to enter the cells), and live cells will show low blue fluorescence, whereas cells undergoing PCD will show an intense blue fluorescence (corresponding to the Hoechst 33342 stain that detects condensed chromatin).

#### 4.6.2. Plasma Membrane Alteration

Evaluation of the plasma membrane integrity was carried out with the SYTOX Green assay (Life Technologies, Madrid, Spain). For the experiment, IC_90_ of the compound was incubated with the cells (5 × 10^5^ cells/mL) for 24 h at 37 °C. Afterwards, the dye was added, and after 15 min of incubation, images were taken using the EVOS™ M5000 Imaging System (Invitrogen by Thermo Fisher Scientific). The SYTOX Green dye shows green fluorescence when binding to the DNA of cells with plasma membrane damage.

#### 4.6.3. Reactive Oxygen Species (ROS) Generation

The CellROX Deep Red fluorescent assay (Thermo Fisher Scientific) was used for the evaluation of ROS generation. Firstly, *Naegleria fowleri* trophozoites (5 × 10^5^ cells/mL) were incubated with the IC_90_ of compound **3** at 37 °C for 24 h. After that, the CellROX dye was added and incubated for 30 min in the same conditions. Finally, red fluorescence was observed in an EVOS™ M5000 Imaging System (Invitrogen by Thermo Fisher Scientific) corresponding to the presence of ROS.

#### 4.6.4. Analysis of Mitochondrial Function Disruption

Two different commercial kits were used for the evaluation of the mitochondrial function condition as described below.

##### Mitochondrial Membrane Potential

The collapse of an electrochemical gradient across the mitochondrial membrane in the treated cells during PCD induction was evaluated with the JC-1 Mitochondrial Membrane Potential Detection Kit (Cayman Chemicals Vitro SA, Madrid, Spain). Amoebae were incubated with the IC_90_ of the evaluated compound for 24 h at 37 °C and the JC-1 dye was added later.

This kit allows to distinguish between two different cell types: live cells will show only red fluorescence as the dye appears as J-aggregate complexes, whereas cells undergoing PCD will show both red and green fluorescence as the JC-1 remains in its monomeric form.

##### Evaluation of ATP Levels

For the ATP level measurement, the CellTiter-Glo Luminescent Cell Viability Assay (PROMEGA BIOTECH IBÉRICA S.L, Madrid, Spain) was used according to the manufacturer’s instructions. The effect of compound **3** on ATP production was evaluated after the incubation of 5 × 10^5^ cells/mL with the previously calculated IC_90_ of the compound.

## 5. Conclusions

In conclusion, (6a*R*, 7*R*, 12a*R*)-2-fluoro-7-phenyl-6a,7,12,12a-tetrahydro-6*H*-chromeno 4,3-*b* 1,8 naphthyridine named compound **3** in this study, showed high activity against the two different *Naegleria fowleri* strains included in this study, inducing PCD processes and showing low toxicity. Therefore, this compound could be a good candidate for future in vivo assays to develop new PAM therapeutic protocols.

## Figures and Tables

**Figure 1 pharmaceuticals-14-01013-f001:**
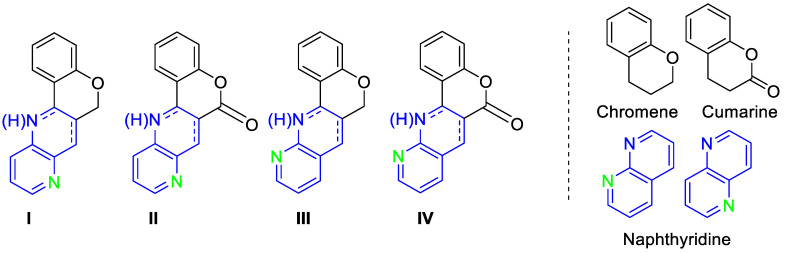
Structure of the hybrid molecules prepared for their biological evaluation.

**Figure 2 pharmaceuticals-14-01013-f002:**
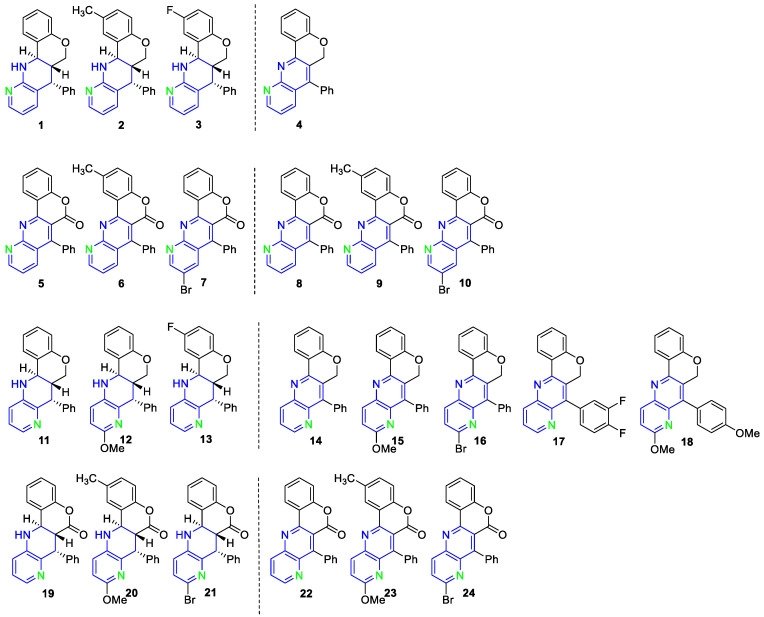
Different families of fused naphthyridines evaluated as antiparasitic agents against *Naegleria fowleri*. Compounds **1**–**4** correspond to 1,8 naphthyridines fused with chromene. Compounds **5**–**10** are 1,8 naphthyridines fused with chromenone rings. Compounds **11**–**18** correspond to 1,5 naphthyridines fused with chromene. Compounds **19**–**24** are 1,5 naphthyridines fused with chromenone.

**Figure 3 pharmaceuticals-14-01013-f003:**
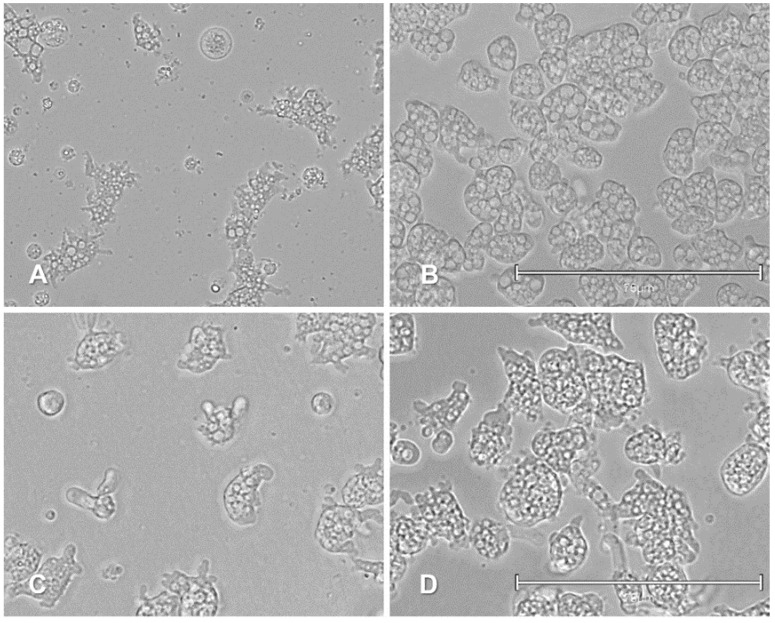
Growth inhibition of *Naegleria fowleri* (ATCC^®^ 30808™) trophozoites at 48 h at IC_50_ concentration. Compound **3** (**A**) and negative control (**B**) with *Naegleria fowleri* (ATCC^®^ 30215™) trophozoites at 48 h at IC_50_ concentration. Compound **3** (**C**) and negative control (**D**). All images (40×) are representative of the cell population of treated amoeba and are based on the EVOS™ M5000 Imaging System, Invitrogen by Thermo Fisher Scientific, Madrid, Spain. Scale bars represent 75 µm.

**Figure 4 pharmaceuticals-14-01013-f004:**
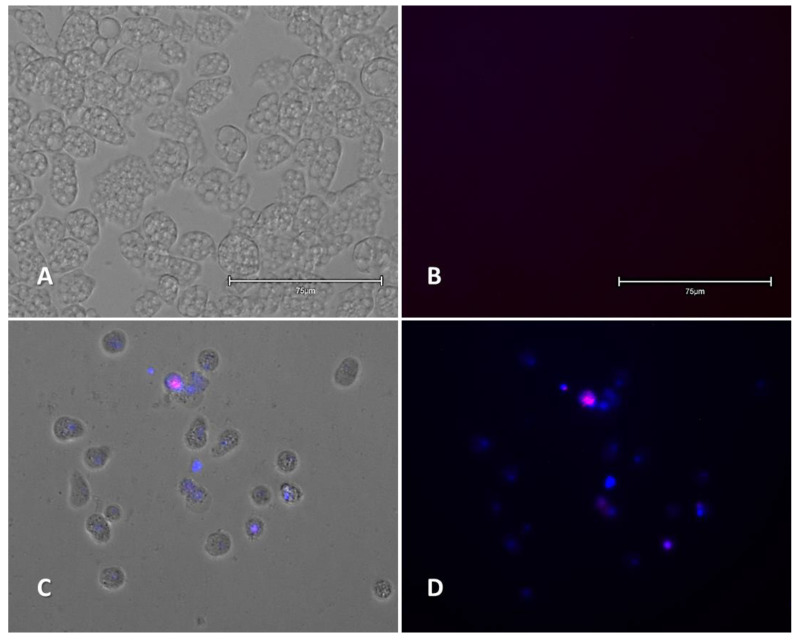
*Naegleria fowleri* (ATCC^®^ 30808™) trophozoites incubated with IC_90_ of the evaluated compound **3** for 24 h. Negative control (**A**,**B**); compound **3** (**C**,**D**). Hoechst/PI stain is different in control cells, where no fluorescence is observed (**B**), and in treated cells, where the nuclei are intense blue and one of them shows red fluorescence inside (**D**). Images (40×) are representative of the cell population observed in the performed experiments. Images were obtained using an EVOS™ M5000 Imaging System, Invitrogen by Thermo Fisher Scientific, Madrid, Spain. Scale bars represent 75 µm.

**Figure 5 pharmaceuticals-14-01013-f005:**
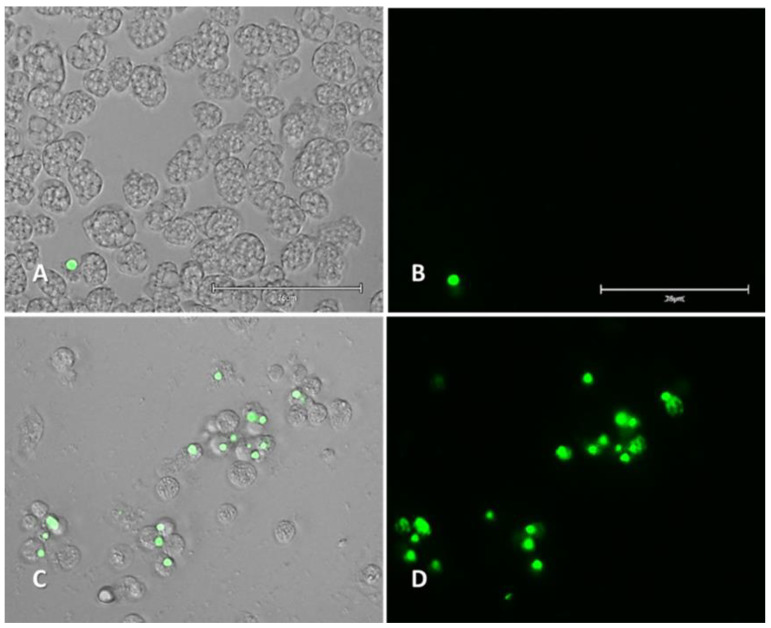
Permeation of the *Naegleria fowleri* (ATCC^®^ 30808™) plasma membrane to the vital dye SYTOX green caused by addition of compound **3** (**C**,**D**) at IC_90_ after 24 h. Negative control (**A**,**B**). Images (40×) are representative of the cell population observed in the performed experiments. Images were obtained using an EVOS™ M5000 Imaging System, Invitrogen by Thermo Fisher Scientific, Madrid, Spain. Scale bars represent 75 µm.

**Figure 6 pharmaceuticals-14-01013-f006:**
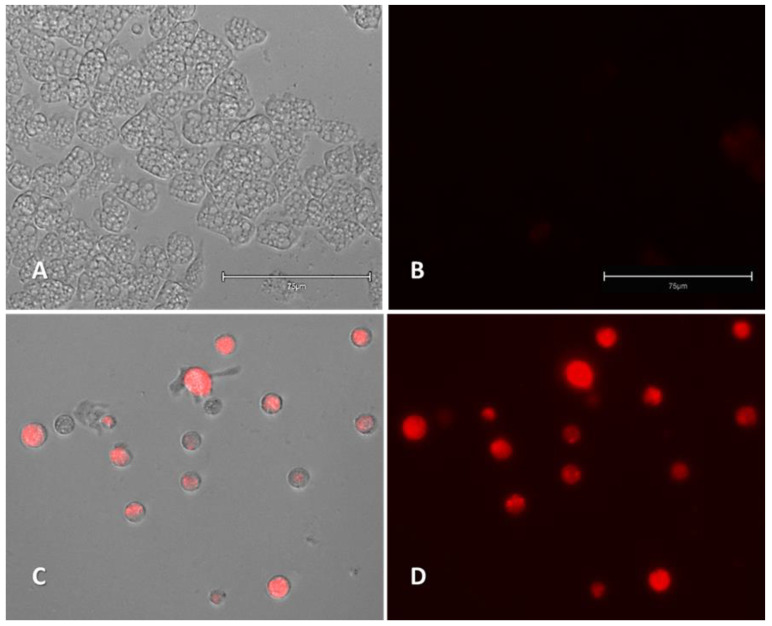
Increase in intracellular ROS levels (red fluoresncence) caused by the addition of compound **3** (**C**,**D**) at IC_90_ after 24 h of incubation with *Naegleria fowleri* (ATCC^®^ 30808™). Negative control (**A**,**B**). The evaluated compound was added to the cells and exposed to CellROX^®^ Deep Red (5 μM, 30 min) at 37 °C in the dark. Images (40×) are representative of the cell population observed in the performed experiments. Images were obtained using an EVOS™ M5000 Imaging System, Invitrogen by Thermo Fisher Scientific, Madrid, Spain. Scale bars represent 75 µm.

**Figure 7 pharmaceuticals-14-01013-f007:**
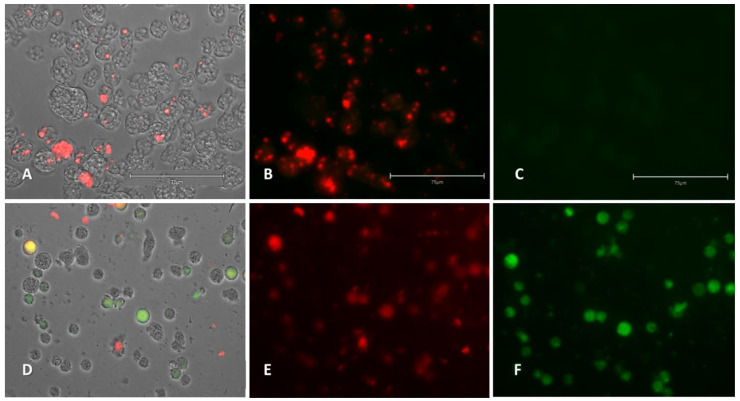
Effect of compound **3** (**D**–**F**) on the mitochondrial membrane potential against *Naegleria fowleri* (ATCC^®^ 30808™). Negative control (**A**–**C**) corresponds to *Naegleria fowleri* (ATCC^®^ 30808™) without treatment. JC-1 dye accumulates in the mitochondria of healthy cells as aggregates and emits red fluorescence (red channel: **B**–**E**). Cells treated with the IC_90_ of the compound for 24 h emitted green fluorescence (green channel: **C**–**F**); due to the decrease in mitochondrial membrane potential, the JC-1 dye remained in the cytoplasm in monomeric form and emitted green fluorescence. Images (40×) are representative of the cell population observed in the performed experiments. Images were obtained using an EVOS™ M5000 Imaging System, Invitrogen by Thermo Fisher Scientific, Madrid, Spain. Scale bars represent 75 µm.

**Figure 8 pharmaceuticals-14-01013-f008:**
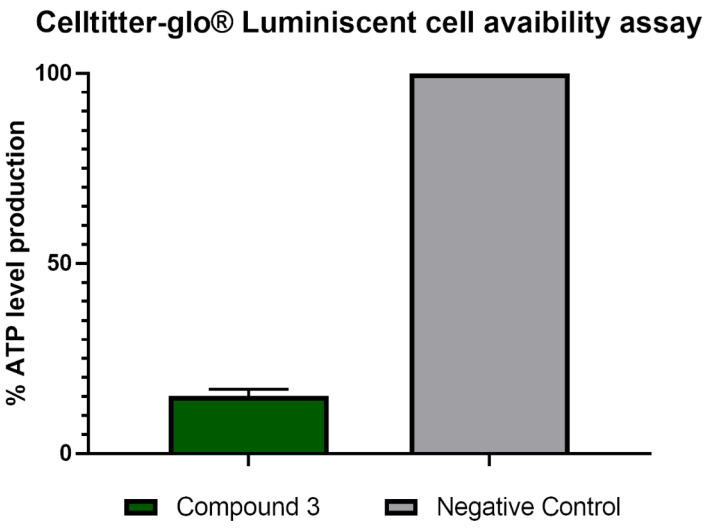
ATP levels in relative percentages compared to the negative control in *Naegleria fowleri* (ATCC^®^ 30808™) trophozoites after 24 h of incubation with IC_90_ of the evaluated compound. Error bars represent the standard deviation (SD). Each data point indicates the average of the results of three measurements. The results show a decrease in the ATP level following treatment with compound **3** of 84.90% compared to the negative control (cells without treatment).

**Table 1 pharmaceuticals-14-01013-t001:** Inhibitory concentrations (IC_50_/IC_90_) of the evaluated compounds against the trophozoite stage of *Naegleria fowleri* type strains ATCC^®^ 30808™ and ATCC^®^ 30215™, and cytotoxicity values (CC_50_) against murine macrophages J774A.1 strain ATCC^®^ TIB-67™. N/D indicates that values were not determinate. The selectivity index (SI) of the active compounds was also calculated.

Compound	IC_50_ (μM) ATCC^®^ 30808	IC_50_ (μM) ATCC^®^ 30215	CC_50_ (μM) ATCC^®^ TIB-67	SI (CC_50_/IC_50_) ATCC^®^ TIB-67/ATCC^®^ 30808	IC_90_ (μM) ATCC^®^ 30808	IC_90_ (μM) ATCC^®^ 30215
1	>100	N/D	N/D	N/D	N/D	N/D
2	>100	N/D	N/D	N/D	N/D	N/D
3	61.45 ± 5.27	76.61 ± 10.84	>300	>4.88	199.02	>200
4	>100	N/D	N/D	N/D	N/D	N/D
5	>100	N/D	N/D	N/D	N/D	N/D
6	>100	N/D	N/D	N/D	N/D	N/D
7	>100	N/D	N/D	N/D	N/D	N/D
8	>100	N/D	N/D	N/D	N/D	N/D
9	>100	N/D	N/D	N/D	N/D	N/D
10	>100	N/D	N/D	N/D	N/D	N/D
11	>100	N/D	N/D	N/D	N/D	N/D
12	>100	N/D	N/D	N/D	N/D	N/D
13	>100	N/D	N/D	N/D	N/D	N/D
14	>100	N/D	N/D	N/D	N/D	N/D
15	>100	N/D	N/D	N/D	N/D	N/D
16	>100	N/D	N/D	N/D	N/D	N/D
17	>100	N/D	N/D	N/D	N/D	N/D
18	>100	N/D	N/D	N/D	N/D	N/D
19	>100	N/D	N/D	N/D	N/D	N/D
20	>100	N/D	N/D	N/D	N/D	N/D
21	>100	N/D	N/D	N/D	N/D	N/D
22	>100	N/D	N/D	N/D	N/D	N/D
23	>100	N/D	N/D	N/D	N/D	N/D
24	>100	N/D	N/D	N/D	N/D	N/D
Amphotericin B	0.12 ± 0.03	0.16 ± 0.02	≥200	≥1652.89	0.35 ± 0.02	0.41 ± 0.12
Miltefosine	38.74 ± 4.23	81.57 ± 7.23	127.89 ± 8.85	3.301	89.47 ± 17.37	>200

## Data Availability

Data is contained within the article and Supplementary Material.

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
