# Peer review of "Naphthyridine Derivatives Induce Programmed Cell Death in Naegleria fowleri"

_pharmaceuticals, 2021, doi:10.3390/ph14101013_

Round 1

Reviewer 1 Report

The manuscript entitled, “Naphthyridine derivatives eliminate Naegleria fowleri by in- 2 duction of Programmed Cell Death” by Rizo-Liendo et al describes the identification of a potential naphthyridine derivative (compound 3) to treat deadly disease caused by the brain-eating ameba Naegleria fowleri. While this study deals with discovery of new drug which is appreciated, however, the use of English language in the manuscript could have been better. Additionally, some of the experimental controls were missing. My suggestions to improve this manuscript are as follows:

  1. English language should be improved.
  2. Is the compound 3 expected to cross the blood-brain-barrier (BBB)? Is there any evidence or reference in favor of this? Drugs that can cross the BBB is a vital criterion for a compound to be effective against primary amebic encephalitis (PAM) caused by N. fowleri?
  3. Figure 2: The Figure Legend is incomplete. Please describe this Figure.
  4. Is there a Figure 3 that was not incorporated in the manuscript? I did not find it.
  5. Two strains of fowleri, ATCC 30808 and ATCC 30215 have been used in this study. There should be some description of these strains. Why did the authors choose these two strains for their study?
  6. Table 1: (a) Why was ATCC 30215 strain not used in the IC90 (µM) determination for compound 3? (b) Similarly, ATCC 30215 strain was not used in the IC50 (µM) or IC90 (µM) determination for Amphotericin B or Miltefosine? Because the origins of ATCC 30808 and ATCC 30215 are highly diverse (Belgium and Australia, respectively), addition of IC50 (µM) and IC90 (µM) values for the other strain would have been important for comparison purposes.
  7. To remove confusions, the derivative names of the naphthyridine can be written in the forms “1,5-“ or “1,8-“ in places of “[1,5]” or “[1,8]”, respectively, that is, without using any brackets. In current forms, they can be confused with the reference numbers.
  8. Figures 4-8: If only the fowleri strain ATCC 30808 has been used in these experiments, this should be mentioned in the respective Figure legends. However, use of both strains would have strengthened the conclusions of these studies. Additionally, Amphotericin B or Miltefosine (or both) should have been used as the positive controls in these experiments.
  9. Line 182: The best genome sequence data for fowleri to-date has been described by Ali et al in a recent article (see below). This article should have been included here. [Ali IKM, Kelley A, Joseph SJ, Park S, Roy S, Jackson J, Cope JR, Rowe LA, Burroughs M, Sheth M, Batra D, Loparev V. Draft Chromosome Sequences of a Clinical Isolate of the Free-Living Ameba Naegleria fowleri. Microbiol Resour Announc. 2021 Apr 15;10(15):e01034-20. doi: 10.1128/MRA.01034-20. PMID: 33858935; PMCID: PMC8050977.]
  10. Discussion: Any or lack of any information regarding the crossing of BBB by compound 3 should be included in the Discussion section. Also, authors should mention that this compound should be further verified in the experimental animal studies such as mouse model.
  11. Conclusions: In line-325, it says, “……………….these compounds…………..”. I don’t see any potential of other compounds other than the compound 3 tested in this study; therefore, it should say ‘this compound’.
  12. Line-25: Add the unit of measurement (i.e., “µM”).
  13. Line-209: What is “TLCs”?
  14. Use the italicized forms of “Naegleria” or “ fowleri” throughout the manuscript. For examples, in lines- 23, 25, 47, 104, 116, etc.

Author Response

The manuscript entitled, “Naphthyridine derivatives eliminate Naegleria fowleri by in- 2 duction of Programmed Cell Death” by Rizo-Liendo et al describes the identification of a potential naphthyridine derivative (compound 3) to treat deadly disease caused by the brain-eating ameba Naegleria fowleri. While this study deals with discovery of new drug which is appreciated, however, the use of English language in the manuscript could have been better. Additionally, some of the experimental controls were missing. My suggestions to improve this manuscript are as follows:

  1. English language should be improved.

Grammar was revised by native speaker.

  1. Is the compound 3 expected to cross the blood-brain-barrier (BBB)? Is there any evidence or reference in favor of this? Drugs that can cross the BBB is a vital criterion for a compound to be effective against primary amebic encephalitis (PAM) caused by N. fowleri?

These type of compounds have been proven to inhibit acetylcholinesterases and have been proposed as treatment of Alzheimer disease, hence we suppose they will be Abel to cross BBB. This information was added in the discussion section and also a statement mentioning that testing of BBB crossing by compounds 3 should be performed in the near future. Eur J Med Chem

. 2014 Feb 12;73:141-52. doi: 10.1016/j.ejmech.2013.12.008. Epub 2013 Dec 18

  1. Figure 2: The Figure Legend is incomplete. Please describe this Figure.

Description included.

  1. Is there a Figure 3 that was not incorporated in the manuscript? I did not find it.

This was a mistake we have modified the numbers.

  1. Two strains of fowleri, ATCC 30808 and ATCC 30215 have been used in this study. There should be some description of these strains. Why did the authors choose these two strains for their study?

Description added.

  1. Table 1: (a) Why was ATCC 30215 strain not used in the IC90 (µM) determination for compound 3? (b) Similarly, ATCC 30215 strain was not used in the IC50 (µM) or IC90 (µM) determination for Amphotericin B or Miltefosine? Because the origins of ATCC 30808 and ATCC 30215 are highly diverse (Belgium and Australia, respectively), addition of IC50 (µM) and IC90 (µM) values for the other strain would have been important for comparison purposes.

These values have been added to table 1

  1. To remove confusions, the derivative names of the naphthyridine can be written in the forms “1,5-“ or “1,8-“ in places of “[1,5]” or “[1,8]”, respectively, that is, without using any brackets. In current forms, they can be confused with the reference numbers.

Modified as suggested

  1. Figures 4-8: If only the fowleri strain ATCC 30808 has been used in these experiments, this should be mentioned in the respective Figure legends. However, use of both strains would have strengthened the conclusions of these studies. Additionally, Amphotericin B or Miltefosine (or both) should have been used as the positive controls in these experiments.

Results using both strains are highly similar, hence no performance of experiments in duplicate. The information was added to the results section. The controls are established by each kit used and are described in the methods section.

  1. Line 182: The best genome sequence data for fowleri to-date has been described by Ali et al in a recent article (see below). This article should have been included here. [Ali IKM, Kelley A, Joseph SJ, Park S, Roy S, Jackson J, Cope JR, Rowe LA, Burroughs M, Sheth M, Batra D, Loparev V. Draft Chromosome Sequences of a Clinical Isolate of the Free-Living Ameba Naegleria fowleri. Microbiol Resour Announc. 2021 Apr 15;10(15):e01034-20. doi: 10.1128/MRA.01034-20. PMID: 33858935; PMCID: PMC8050977.]

Incuded as suggested

  1. Discussion: Any or lack of any information regarding the crossing of BBB by compound 3 should be included in the Discussion section. Also, authors should mention that this compound should be further verified in the experimental animal studies such as mouse model.

We have included a statement about this issue in the discussion section: “Moreover, naphthyridines and their derivatives have been previously reported to in-hibit acetylcholinesterases and have been proposed as interesting anti-Alzheimer lead compounds. Hence, these molecules should be able to cross the brain blood barrier sup-porting their potential as anti-PAM agents. However, BBB crossing by these compounds and in vivo efficacy should be further verified in the experimental animal studies in the near future”

  1. Conclusions: In line-325, it says, “……………….these compounds…………..”. I don’t see any potential of other compounds other than the compound 3 tested in this study; therefore, it should say ‘this compound’.

Modified as suggested

  1. Line-25: Add the unit of measurement (i.e., “µM”).

Modified as suggested

  1. Line-209: What is “TLCs”?

Thin layer chromatography (TLC)

  1. Use the italicized forms of “Naegleria” or “ fowleri” throughout the manuscript. For examples, in lines- 23, 25, 47, 104, 116, etc.

Corrected.

Reviewer 2 Report

This manuscript reports on the promising anti-amoebal activity of a novel drug synthesised by the authors. They determine that not only does this drug kill the amoebae, but it does so through the induction of PCD.  The authors have a record of high-quality work in the area of protistan PCD and the paper is well written and clear.

The title “Naphthyridine derivatives eliminate Naegleria fowleri by induction of Programmed Cell Death.” Might suggest that the drug eliminates the amoeba from a host.  Hopefully later studies will show that it does but perhaps “Naphthyridine derivatives induce Programmed Cell Death in Naegleria fowleri.” Is a better title?

Throughout the manuscript Naegleria fowleri and N. fowleri are sometime italicized but often not. These terms should all be in italics.

Line 19 “risky” rather than “risks”

Line 20 “currently“ rather than “until present”

Line 25 need to mention IC50 and IC90 here and to give the units (µM)

Line 27 Also needs units (µM)

Line 28 “damage to” rather than “damages at”

Line 28 A comma is needed after ATP levels.

Line 35 “the free-living amoeba parasite” rather than “free-living amoebae parasite”

Line 38 needs “the” after “known as”. Also “species” not “specie”

Line 44 the word “and” is better here than “nevertheless”

Reference 3 is not a direct reference for this figure of 440 cases, it just cites other papers, also reference 4 is a URL which should be avoided as URLs often prove to be temporary. There are two papers which contain estimates of case numbers are given doi.org/10.1093/cid/ciaa520 and doi.org/10.1016/j.pt.2019.10.008. It would be better to cite these.

Line 45 “performance” rather than ”performing”.

Line 62 remove “On the one hand,”

Line 70 remove “on the other hand”

Line 106 needs units (µM)

Line 178. It is certainly true that PCD rather than necrosis results in less inflammation in the cells of mammals, but it is much less certain if death by PCD in Naegleria results in less inflammation than would be caused by death by necrosis of Naegleria within a patient. I do not think that this has been determined experimentally.

Line 325.  This should be “this compound” not “these compounds” as only compound 31 has activity.

Author Response

This manuscript reports on the promising anti-amoebal activity of a novel drug synthesised by the authors. They determine that not only does this drug kill the amoebae, but it does so through the induction of PCD.  The authors have a record of high-quality work in the area of protistan PCD and the paper is well written and clear.

The title “Naphthyridine derivatives eliminate Naegleria fowleri by induction of Programmed Cell Death.” Might suggest that the drug eliminates the amoeba from a host.  Hopefully later studies will show that it does but perhaps “Naphthyridine derivatives induce Programmed Cell Death in Naegleria fowleri.” Is a better title?

Modified as suggested

Throughout the manuscript Naegleria fowleri and N. fowleri are sometime italicized but often not. These terms should all be in italics.

Corrected.

Line 19 “risky” rather than “risks”

Modified as suggested

Line 20 “currently“ rather than “until present”

Modified as suggested

Line 25 need to mention IC50 and IC90 here and to give the units (µM)

Modified as suggested

Line 27 Also needs units (µM)

Modified as suggested

Line 28 “damage to” rather than “damages at”

Modified as suggested

Line 28 A comma is needed after ATP levels.

Modified as suggested

Line 35 “the free-living amoeba parasite” rather than “free-living amoebae parasite”

Modified as suggested

Line 38 needs “the” after “known as”. Also “species” not “specie”

Modified as suggested

Line 44 the word “and” is better here than “nevertheless”

Modified as suggested

Reference 3 is not a direct reference for this figure of 440 cases, it just cites other papers, also reference 4 is a URL which should be avoided as URLs often prove to be temporary. There are two papers which contain estimates of case numbers are given doi.org/10.1093/cid/ciaa520 and doi.org/10.1016/j.pt.2019.10.008. It would be better to cite these.

Modified as suggested

Line 45 “performance” rather than ”performing”.

Modified as suggested

Line 62 remove “On the one hand,”

Modified as suggested

Line 70 remove “on the other hand”

Modified as suggested

Line 106 needs units (µM)

Modified as suggested

Line 178. It is certainly true that PCD rather than necrosis results in less inflammation in the cells of mammals, but it is much less certain if death by PCD in Naegleria results in less inflammation than would be caused by death by necrosis of Naegleria within a patient. I do not think that this has been determined experimentally.

It has not been determined experimentally but it is expected and desirable as it has been suggested by our labs and others in previous reports and also by other scientists working on therapeutics.

Line 325.  This should be “this compound” not “these compounds” as only compound 31 has activity.

Modified as suggested

Reviewer 3 Report

Dear authors,

This work is interesting and useful for the scientific community working on Naegleria Fowleri and PAM. This work is well structured and presented. However, changes must be incorporated before publication.

1) In the material and methods, you indicated to have used and cultivated the strain ATCC 30808 Naegleria fowleri, however in table 1, you present the result of the compounds used on two different strains (ATCC® 30215 and ATCC 30808), clarify and notify it in the manuscript.

2) Why did you use these two strains (ATCC® 30215 and ATCC 30808) and not the others? is one more pathogenic than the other? have they been isolated from a human or other sample? have it already been used in other studies? after how many generations (culture) has it been used? add these information in manuscript please.

3) On line 23, line 105, 116...: Please write N. fowleri in italics.

4) Figure 2: complete the legend, explain what can be seen in each of the 4 photos.

5) Fig 7: which negative control is used (A-B-C), indicate it in the legend please.

6) Line 165: Be careful to integrate the bibliography into the journal formats

"different cancer cell lines 36,37,42 ....."

7) Line 210-221: mention the reference of all the products mentioned and used.

8) Line 236: one extra space please correct this “Briefly, 50   1 of Naegleria foweri cells”.

9) Line 323: Put the conclusion at the end of the discussion and not of the material and methods part. And specify the name of composer 3 in the conclusion, "compound 3 show...."

10) You have specified in the abstract what PCD means, but this must also be described a first time in the main text and you can then use the abbreviations.

Author Response

Dear authors,

This work is interesting and useful for the scientific community working on Naegleria Fowleri and PAM. This work is well structured and presented. However, changes must be incorporated before publication.

  • In the material and methods, you indicated to have used and cultivated the strain ATCC 30808 Naegleria fowleri, however in table 1, you present the result of the compounds used on two different strains (ATCC® 30215 and ATCC 30808), clarify and notify it in the manuscript.

For the activity assays we have used both strains. In the case of PCD studies we only used one since we have seen that the results are similar using any of them. This information was added in the results section.

2) Why did you use these two strains (ATCC® 30215 and ATCC 30808) and not the others? is one more pathogenic than the other? have they been isolated from a human or other sample? have it already been used in other studies? after how many generations (culture) has it been used? add these information in manuscript please.

In material and methods section: Two American Type Culture Collection ATCC 30808 and ATCC 30215 strains of Naegleria fowleri were used to perform the in vitro evaluation of the compounds (LG Promochem, Barcelona, Spain). Amoebas were axenically grown at 37°C in 2% (v/w) Bactocasitone medium (Termo Fisher Scientifc, Madrid, Spain), supplemented with 10% (v/v) fetal bovine serum (FBS) (Biowest, VWR, Spain), 0.3 μg/ml of Penicillin G Sodium Salt (Sigma-Aldrich, Madrid, Spain) and 0.5 mg/ml of Streptomycin sulphate (Sigma-Aldrich, Madrid, Spain). Naegleria fowleri cultures were conserved in a biological security facility of level 3 at the Instituto Universitario de Enfermedades Tropicales y Salud Pública de Canarias following the Spanish goverments´ biosafety guidelines for this pathogen with the minimum number of passages possible to avoid loss of pathogenicity.

3.-On line 23, line 105, 116...: Please write N. fowleri in italics.

Modified as suggested

4) Figure 2: complete the legend, explain what can be seen in each of the 4 photos.

Modified as suggested

5) Fig 7: which negative control is used (A-B-C), indicate it in the legend please.

Modified as suggested.

6) Line 165: Be careful to integrate the bibliography into the journal formats

"different cancer cell lines 36,37,42 ....."

Modified as suggested

7) Line 210-221: mention the reference of all the products mentioned and used.

Modified as suggested

8) Line 236: one extra space please correct this “Briefly, 50   1 of Naegleria foweri cells”.

Modified as suggested

9) Line 323: Put the conclusion at the end of the discussion and not of the material and methods part. And specify the name of composer 3 in the conclusion, "compound 3 show...."

Modified as suggested. However the order of the journal formal requires conclusions after the material and methods section.

10) You have specified in the abstract what PCD means, but this must also be described a first time in the main text and you can then use the abbreviations.

Modified as suggested

Round 2

Reviewer 1 Report

Few minor changes are still needed:

  1. Line-44: Change "report" to 'reported'.
  2. Line-45: Change "above" to 'more than'.
  3. Lines 45-47: Based on the information provided here, Pakistan had more PAM cases than the USA. The statement needs correction.
  4. Table 1: ">" signs have been used in some of the values. It is better to use the exact values, when possible.
  5. Figure 2 legend: The modified Figure legend has been truncated in the PDF version of the revised manuscript.  
  6. Figures 5 and 7 legends are missing, not shown in the revised PDF version.

Author Response

  1. Line-44: Change "report" to 'reported'--> Modified.
  2. Line-45: Change "above" to 'more than'.--> Corrected
  3. Lines 45-47: Based on the information provided here, Pakistan had more PAM cases than the USA. The statement needs correction. --> There was a mistake in the data which have been corrected now.
  4. Table 1: ">" signs have been used in some of the values. It is better to use the exact values, when possible.--> As stated by the reviewer sometimes it is not possible which is the case in the data mentioned from this table.
  5. Figure 2 legend: The modified Figure legend has been truncated in the PDF version of the revised manuscript.  --> Corrected.
  6. Figures 5 and 7 legends are missing, not shown in the revised PDF version --> Corrected.